# A scoping review of the post-discharge care needs of babies requiring surgery in the first year of life

**Francesca Giulia Maraschin**◉*, **Fidelis Jacklyn Adella**◉, **Shobhana Nagraj**◉

Health Systems Collaborative, Centre for Global Health Research, The Nuffield Department of Medicine, The University of Oxford, Oxford, United Kingdom

* maraschinfran@gmail.com

**Data Availability Statement:** This scoping review did not generate any new primary data. The study is based entirely on previously published research articles, reports, and other resources. All references to these sources are included within the

## Abstract

Congenital anomalies are among the leading causes of under-5 mortality, predominantly impacting low- and middle-income countries (LMICs). A particularly vulnerable group are babies with congenital disorders requiring surgery in their first year. Addressing this is crucial to meet SDG-3, necessitating targeted efforts. Post-discharge, these infants have various care needs provided by caregivers, yet literature on these needs is scant. Our scoping review aimed to identify the complex care needs of babies post-surgery for critical congenital cardiac conditions and non-cardiac conditions. Employing the Joanna Briggs Institute's methodological framework for scoping reviews we searched Pubmed, EMBASE, CINAHL, PsychINFO, and Web of Science databases. Search terms included i) specific congenital conditions (informed by the literature and surgeons in the field), ii) post-discharge care, and iii) newborns/infants. English papers published between 2002–2022 were included. Findings were summarised using a narrative synthesis. Searches yielded a total of 10,278 papers, with 40 meeting inclusion criteria. 80% of studies were conducted in High-Income Countries (HICs). Complex care needs were shared between cardiac and non-cardiac congenital conditions. Major themes identified included 1. Monitoring, 2. Feeding, and 3. Specific care needs. Sub-themes included monitoring (oxygen, weight, oral intake), additional supervision, general feeding, assistive feeding, condition-specific practices e.g., stoma care, and general care. The post-discharge period poses a challenge for caregivers of babies requiring surgery within the first year of life. This is particularly the case for caregivers in LMICs where access to surgical care is challenging and imposes a financial burden. Parents need to be prepared to manage feeding, monitoring, and specific care needs for their infants before hospital discharge and require subsequent support in the community. Despite the burden of congenital anomalies occurring in LMICs, most of the literature is HIC-based. More research of this nature is essential to guide families caring for their infants post-surgical care.

manuscript. Any inquiries related to these sources should be directed to the original authors or publishers.

**Funding:** The authors received no specific funding for this work.

**Competing interests:** The authors have declared that no competing interests exist.

## Introduction

Sustainable Development Goal 3.2 (SDG) is to reduce under-5 mortality to 25 deaths per 1,000 live births [1]. Congenital anomalies are amongst the top five causes of death in children under 5, accounting for 240,000 newborn deaths and a loss of 25.3–38.8 million disability adjusted life years (DALYs) each year [2, 3]. Most of these birth defects (94%) occur in low- and middle-income countries (LMICs) [3], likely attributable to higher fertility rates, increased nutritional deficiencies in pregnancy, intrauterine infection, higher teratogen exposure and decreased access to timely obstetric care [3]. Given that congenital defects contribute significantly to under-5 mortality, targeting survival of infants with congenital defects could improve neonatal mortality globally.

Congenital heart disease (CHD) is a leading cause of infant death, with non-cardiac anomalies also contributing [4]. Several of these conditions require surgery for survival within the first year of life and pose problems for families in low-resource settings, who may lack access to specialist surgical services, face catastrophic health expenditure because of their babies requiring surgery, and have additional challenges when caring for babies after discharge from hospital. Among these babies with complex care needs, are those born with critical congenital heart defects, classified by the American Heart Association [5], and those with common non-cardiac congenital anomalies requiring surgery for survival in the first year of life [6]. These conditions are outlined in Table 1.

Surgical care for babies (neonatal surgery) has made dramatic improvements in the last 50 years, meaning many surgical neonates survive to discharge [7]. Caregivers are expected to meet the needs of these infants which is time-consuming and imposes a financial and emotional burden, especially for caregivers in LMICs, who may have several conflicting demands on time and resources. Adequate care post-discharge is essential for neonatal survival post-surgical care. Despite this, there is little literature exploring the complex care needs of these newborns post-discharge and the implications for caregivers, and those living in resource-limited settings globally.

The main objective of our scoping review was to map literature on the complex care needs of newborns post-discharge for common congenital abnormalities requiring surgery within the first year of life, which, if left untreated would contribute to under-five mortality. Traditionally, cardiac, and non-cardiac anomalies would be managed separately, but in practice, especially in LMICs, hospitals that care for these babies deal with both cardiac and non-cardiac anomalies, hence our including both in this review. The aim was to highlight the array of care needs for these babies and the potential impact on caregivers. We also aimed to identify any gaps in the literature concerning post-discharge care of the surgical neonate to inform further research.

## Methods

We conducted a scoping review using methodology informed by the Joanna Briggs Institute's methodology for scoping reviews [8].

### Search strategy

We searched PubMed, EMBASE, PsychINFO, CINAHL and Web of Science Core Collection databases with a search strategy developed with a University Librarian. The most recent search was done on October 18, 2022. The choice of common conditions requiring surgery for survival within the first year of life (Table 1) was informed by the scientific literature on neonatal surgery [9–12], and in consultation with clinicians working in neonatal surgery, cardiac surgery and neonatology in LMIC settings. We used terms relating to i) each aforementioned

**Table 1. Description of each critical congenital heart defect and common non-cardiac congenital anomalies requiring surgery in the first year of life.**

| Condition | Description |
|---|---|
| **Critical congenital heart defects** | |
| Hypoplastic Left Heart Syndrome (HLHS) | A rare congenital heart defect where the left side of the heart is underdeveloped, affecting blood flow to the body. |
| Pulmonary Atresia | A condition where the pulmonary valve, which controls blood flow from the heart to the lungs, doesn't form properly or is blocked. |
| Tetralogy of Fallot (TOF) | A combination of four heart defects that affect the structure of the heart, leading to oxygen-poor blood being pumped into the body. |
| Total Anomalous Pulmonary Venous Return (TAPVR) | An abnormality where the pulmonary veins that bring oxygenated blood from the lungs to the heart connect to the wrong part of the heart or other blood vessels. |
| Transposition of the Great Arteries | A condition where the two main arteries of the heart are switched, resulting in oxygen-poor blood being circulated to the body and oxygen-rich blood to the lungs. |
| Tricuspid Atresia | A rare defect where the tricuspid valve, which controls blood flow between the right atrium and right ventricle of the heart, is missing or abnormally developed. |
| Truncus Arteriosus | A condition where a single blood vessel arises from the heart, instead of separate vessels for the lungs and the body, leading to mixing of oxygenated and deoxygenated blood. |
| **Non-cardiac congenital anomalies** | |
| Congenital Diaphragmatic Hernia (CDH) | A birth defect where there is an opening in the diaphragm, allowing organs from the abdomen to move into the chest and potentially affecting lung development. |
| Anorectal Malformation (ARM) | An abnormality in the development of the rectum and anus, which can result in improper formation of the anal opening and difficulties in passing stool. |
| Oesophageal Atresia with/without Tracheoesophageal Fistula | A condition where the upper oesophagus does not connect with the lower oesophagus and stomach, with or without an abnormal connection (fistula) between the oesophagus and trachea (tracheoesophageal fistula). |
| Gastroschisis | A birth defect where the abdominal wall doesn't close properly, causing the intestines or other organs to protrude outside the body through a hole beside the belly button. |

congenital condition, AND ii) post-discharge care AND iii) newborns/infants (see S1 Table). Papers were included if they were published in the last twenty years (2002–2022), in the English language, discussed the discharge needs for newborns (0–28 days) or infants (≤1 year). Only papers discussing one of the pre-defined congenital conditions were included. Case reports, case series, abstracts, opinion pieces, textbooks, letters to editors and theses were excluded.

## Screening and selection

Screening of titles and abstracts was conducted by two independent reviewers (FM & JA) using *Rayyan* software. This initial screening was blinded and conflicts arising were reviewed by a third reviewer (SN). Full texts of selected papers were reviewed, and final selection of papers meeting inclusion criteria, completed. We supplemented the search with forwards and backwards citation searching, and review of references.

## Data extraction and analysis

A data extraction table was developed a priori and iteratively adapted (see S2 Table). Data extraction was conducted by two independent reviewers (FM & JA), and complex needs

categorised for both cardiac and non-cardiac conditions. We identified recurrent emerging themes within these data using a thematic content analysis and conducted a narrative synthesis of the results.

### Patient and public involvement

No patients were involved in this study.

## Results

The database searches retrieved a total of 13,740 papers, of which 3,462 were duplicates. After screening 10,278 titles, 841 abstracts were reviewed and 808 excluded, as they did not meet the inclusion criteria and 33 papers were included. Searching the references of these papers yielded an additional seven papers, giving a total of 40 papers for final inclusion (Fig 1) [13].

### Study characteristics

Of the 40 papers included in the review, 15 (38%) discussed cardiac conditions and 25 (63%) discussed non-cardiac conditions. Twenty-eight (70%) studies were conducted in the last 10 years (between 2013–2022), 15 studies (38%) were retrospective reviews, with qualitative and cross-sectional studies being the second most common study design. Thirty-two studies (80%) were conducted in high-incomes countries (HICs) with a majority in the USA (48%)—Table 2.

### Complex needs identified

Based on our narrative synthesis, we identified three main themes, and 10 sub-themes, relating to the complex care for the surgical neonate post-discharge. Main themes were: 1. Monitoring,

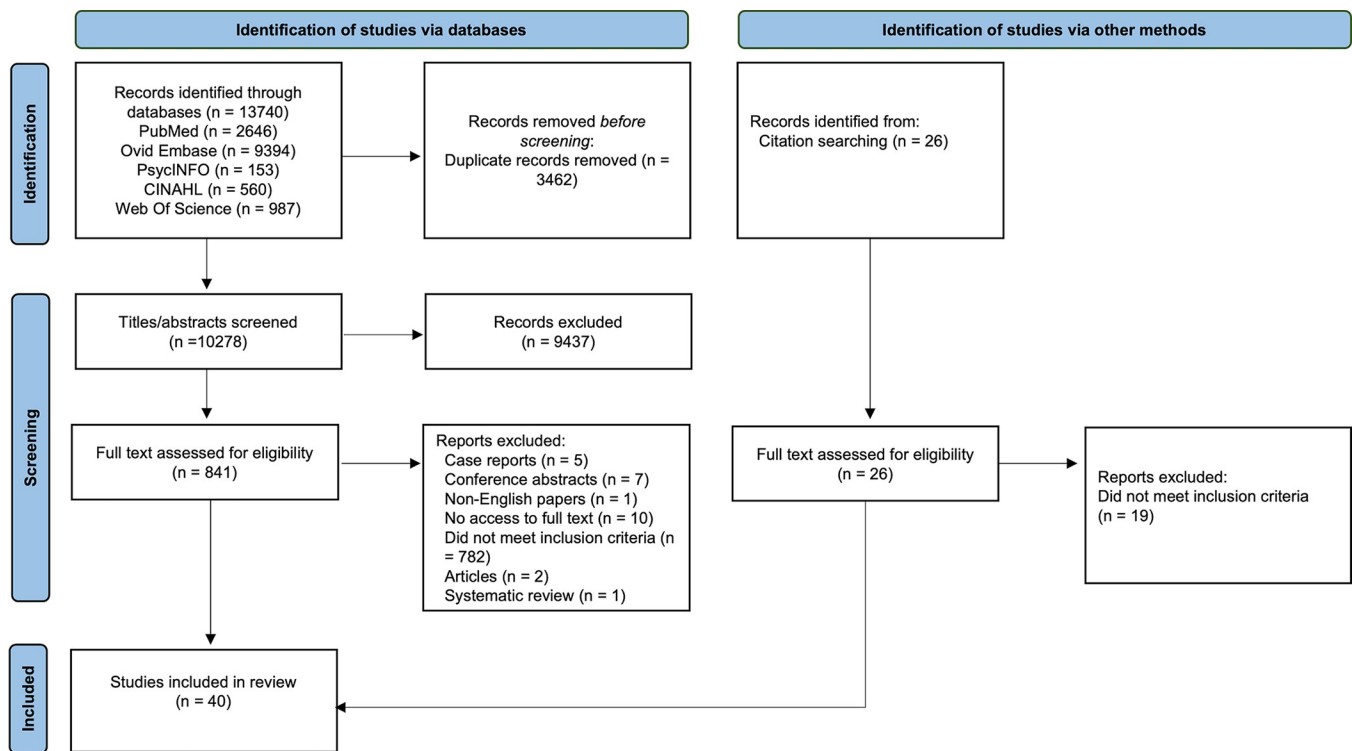

**Fig 1. PRISMA diagram.** PRISMA Diagram showing flow of identification, screening, and inclusion of studies in review.

**Table 2. Study characteristics of included papers.**

| Characteristics | Cardiac (n = 15) | Non-cardiac (n = 25) | Total (n = 40) |
|---|---|---|---|
| **Publication year** | | | |
| 2002–2012 | 4 (27%) | 8 (32%) | 12 (30%) |
| 2013–2022 | 11 (73%) | 17 (68%) | 28 (70%) |
| **Study type** | | | |
| Descriptive | 0 (0%) | 1 (4%) | 1 (3%) |
| Prospective randomised control | 0 (0%) | 1 (4%) | 1 (3%) |
| Cross-sectional | 1 (7%) | 5 (20%) | 6 (15%) |
| Retrospective review | 4 (27%) | 11 (44%) | 15 (38%) |
| Prospective cohort | 1 (7%) | 2 (8%) | 3 (8%) |
| Mixed methods | 1 (7%) | 2 (8%) | 3 (8%) |
| Clinical trial | 0 (0%) | 1 (4%) | 1 (3%) |
| Retrospective cohort | 1 (7%) | 1 (4%) | 2 (5%) |
| Observational | 0 (0%) | 1 (4%) | 1 (3%) |
| Qualitative | 6 (40%) | 0 (0%) | 6 (15%) |
| Prospective interventional | 1 (7%) | 0 (0%) | 1 (3%) |
| **Location** | | | |
| USA | 9 (60%) | 10 (40%) | 19 (48%) |
| UK | 4 (27%) | 1 (4%) | 5 (13%) |
| Canada | 1 (7%) | 3 (12%) | 4 (10%) |
| Germany | 1 (7%) | 1 (4%) | 2 (5%) |
| Nigeria | 0 (0%) | 3 (12%) | 3 (8%) |
| The Netherlands | 0 (0%) | 2 (8%) | 2 (5%) |
| Other | 1 (7%) | 5 (20%) | 6 (15%) |
| **HIC/LMIC** | | | |
| HIC | 15 (100%) | 17 (68%) | 32 (80%) |
| LMIC | 0 (0%) | 7 (28%) | 7 (18%) |

2. Feeding and 3. Specific care needs (Table 3). There was considerable overlap of these needs between cardiac and non-cardiac conditions especially for feeding and monitoring. Figs 2 and 3 detail the individual complex needs of cardiac and non-cardiac articles.

## Monitoring

The need for extra attention and supervision of babies after surgery was mentioned as an important care requirement for caregivers looking after infants post-discharge. This included monitoring babies at home for any signs of worsening illness (such as reduced oxygen levels, worsening breathing, or lethargy), or having to provide extra attention and "cheering up" their infants [14]. Many studies described that caregivers felt like they needed to be hyper-vigilant to notice signs that indicated their baby's condition was worsening. This led to parental anxiety and a lack of sleep. One parent reported that she "*stayed up [to] watch [her child] sleep*" in case the infant's condition worsened [15].

Monitoring of oxygen levels (oxygen saturation) and weight monitoring were mentioned in the context of one particular heart defect known as Hypoplastic Left Heart Syndrome (HLHS), which requires a number of staged operations over the first year of life. Monitoring is required between surgical procedures. Many caregivers were told to measure oxygen saturation and weight daily. For some, keeping the oxygen saturation probe on either the baby's toe or ear lobe, all day, helped ease anxiety because low oxygen levels could be quickly identified.

**Table 3. Themes and sub-themes of complex needs identified.**

| Theme | Sub-theme |
|---|---|
| **Theme 1: Monitoring** | • Monitoring of oxygen levels (oxygen saturation)<br>• Weight monitoring<br>• Oral intake monitoring<br>• Extra attention/supervision (monitoring for signs and symptoms of decompensation, breathing, vomiting etc.) |
| **Theme 2: Feeding** | • General oral feeding practices (having to bottle or breast feed regularly, charting the timing of feeds, calculating volume of feeds, managing vomiting, ensuring adequate food intake, managing non-oral e.g., intravenous (parenteral) nutrition)<br>• Managing feeding techniques that involve assistive feeding devices (NGT[1], gastrostomy/enterostomy tube) |
| **Theme 3: Specific care needs** | • Procedures for anorectal anomalies (anal dilatations)<br>• Care of stoma bag (a stoma is when the colon/intestine is brought to the surface of the abdomen as a hole and covered with a bag to collect colonic/intestinal contents)<br>• Provision of supplemental oxygen<br>• General (giving enemas, changing diapers, washing baby more often, and wound care, protection from infection) |

1 NGT, Nasogastric tube (a thin, flexible tube inserted through the nose and into the stomach to provide food to the baby)

Outcomes tended to be better for those where an interstage home monitoring program was implemented and in one study mortality was 0% in the intervention group compared to 15.8% in the control group (p = 0.039) [16]. Oxygen saturation monitoring was not mentioned as a complex care need for non-cardiac conditions.

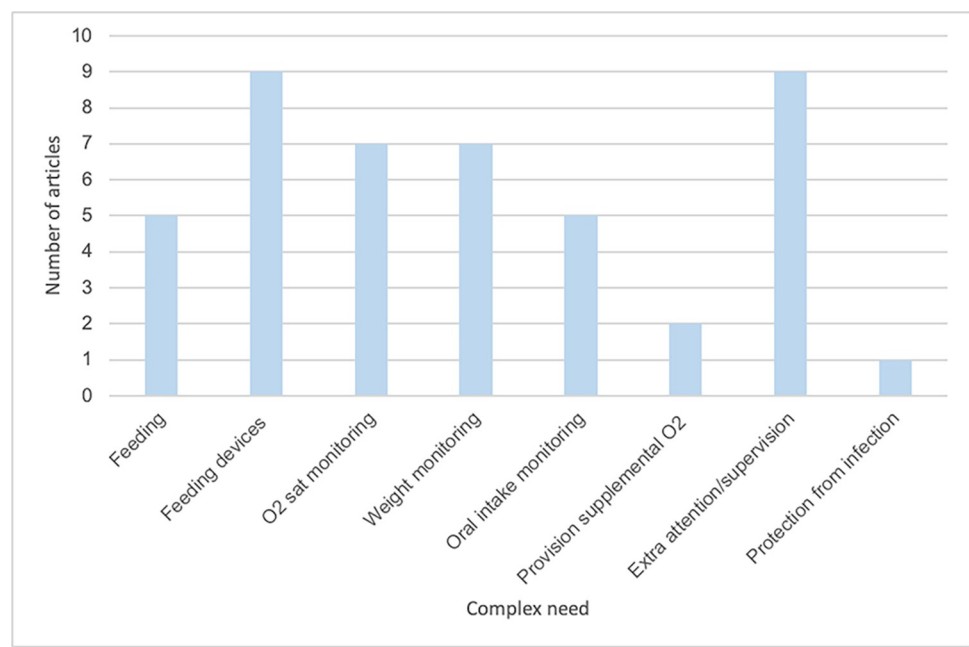

**Fig 2. Bar graph of complex needs in cardiac congenital conditions.** This graph represents the complex needs mentioned in papers discussing cardiac congenital conditions and the number of articles in the review that discussed each of these complex needs.

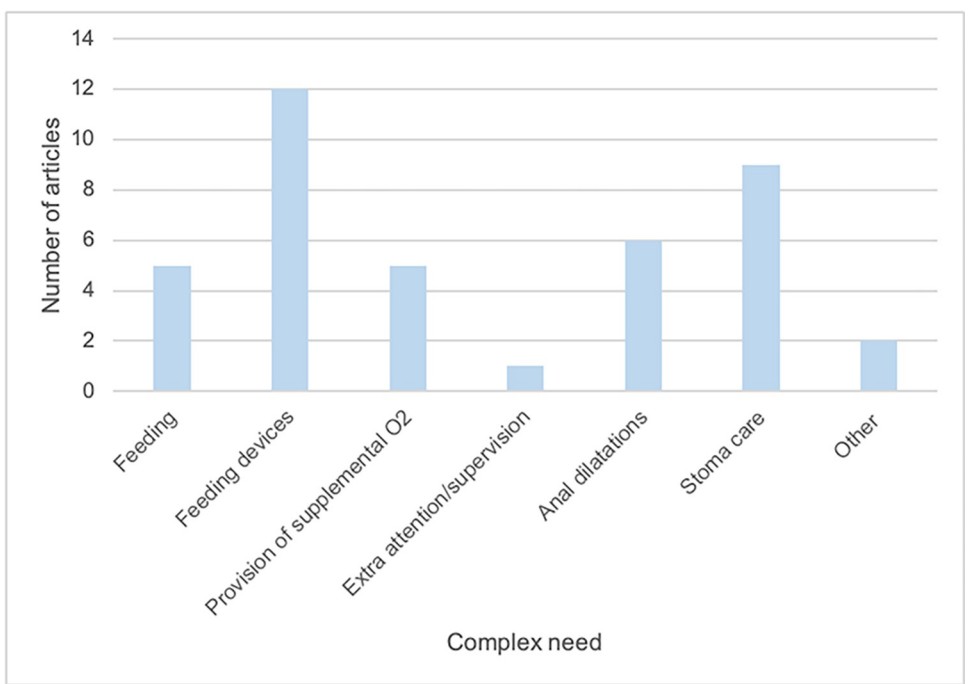

**Fig 3. Bar graph of complex needs in non-cardiac congenital conditions.** This graph represents the complex needs mentioned in papers discussing non-cardiac congenital conditions and the number of articles in the review that discussed each of these complex needs.

## Feeding

Feeding needs included having to feed the baby regularly, chart feeds, manage vomiting post-feeds, calculate feed volume, and manage intravenous (parenteral) nutrition. In one study, as many as 80% of parents reported difficulties with feeding [17]. Some babies with congenital anomalies were unable to be fed orally. These babies required feeding through assistive devices, including nasogastric tubes (NGT), tube-feeding into the stomach (gastrostomy) or feeding into the intestine (enterostomy). Managing such assistive feeding was a complex need mentioned across both cardiac and non-cardiac studies. Rates of tube-fed infants on discharge ranged from 18%-89% [16, 18–29]. Parents were responsible for managing the equipment required for tube feeding, and in many cases, re-siting dislodged tubes, which was identified as a major cause of caregiver anxiety.

## Specific care needs

Provision of oxygen supplementation therapy for babies after surgery was a care need mentioned for babies requiring both cardiac and non-cardiac surgery. Specific at-home needs regarding oxygen supplementation were not mentioned. Instead, studies referred to the percentage of infants discharged on oxygen ranging from 11%-19% [16, 22, 24–26, 30].

Specific care needs mainly related to babies requiring bowel surgery including those born with ARM requiring anal dilatations at home after discharge, and those babies requiring stoma care. Two thirds of studies (n = 7 of 9 total studies) discussing stoma care were conducted in low-resource settings. Solutions for limited access to stoma bags was explored, as well as the acceptability of stomas and barriers to adequate stoma care at home. Krois et al. estimate the cost of stoma care to be $86 USD per month [31]. Many caregivers in these contexts, lack

access to stoma bags and were using solutions such as diapers or cut cloth to collect intestinal/colonic effluent. Osifo et al. and Ameh et al. found that 21.7% and 23% of caregivers respectively had "poor acceptance" of the stoma [32, 33].

Majority of studies regarding the need for caregivers to conduct anal dilatations following surgery for ARM, explored whether dilatations were necessary to reduce post-operative strictures. In the papers included in the review, dilations were prescribed at least daily, and in some cases twice daily upon discharge [34].

In one paper discussing two non-cardiac conditions requiring surgery (ARM and CDH), additional care needs mentioned included wound care, giving enemas to babies, changing diapers, and washing the babies more often [14]. Protection from infection was mentioned by participants in March et al's (2018) study discussing the experiences of parents on transition home after the Norwood procedure for HLHS [35]. Caregivers in this study mentioned not being able to leave home or have visitors to prevent infecting their babies, who had weakened immune systems.

## Discussion

Our scoping review has highlighted that infants requiring surgery in the first year of life have complex needs that can be summarised in to three main themes that share similarities for both cardiac and non-cardiac conditions: monitoring, feeding, and specific care needs.

### Complex care needs

Monitoring was an important theme that emerged from our review. Hyper-vigilance and anxiety were reported by caregivers who feel they need constantly to be on the lookout for signs of deterioration in their infants. March et al. also found that monitoring was a stressor for caregivers, and that anxiety was exacerbated by parents feeling they weren't taught the necessary skills to care for their infants post-discharge and lacked adequate support [35]. In their study measuring psychological function in parents of children with congenital heart disease (CHD)–Doherty at al. (2009) found that elevated levels of psychological distress were apparent in 1/3 of mothers and 1/5 of fathers [36].

Our results showed that feeding was a common complex care need causing considerable anxiety for caregivers of post-surgical babies. March *et al.* similarly found that feeding and managing feeding tubes was a major stressor for parents [35]. Parental anxiety has been shown to increase feeding problems in newborns, and given adequate nutrition is essential for brain development in the first 1000 days of life, parents need to be adequately educated and supported to meet this need [37–39].

While there was considerable overlap in the complex care needs required for both cardiac and non-cardiac pathologies, there were conditions with specific care needs. For example, stoma care and anal dilatations were identified as unique care needs of babies born with ARM. Our review revealed that there were many barriers to stoma care in low-resource settings. This included limited access to stoma bags, and challenges maintaining hygiene and skin care around the stoma site, and poor acceptance of the stoma in these settings.

Given low-resourced healthcare facilities are inundated with surgical cases and might be at a considerable distance for families living in rural areas, for these babies and their families, returning to hospital following initial surgery to reverse the stoma is often delayed, and follow-up care for the baby in the community is challenging–often extending the period of stoma care beyond what would be acceptable in HICs [40]. These issues are also echoed in the wider literature which shows that the lack of stoma therapists, equipment for stoma care, poor compliance and low acceptance rates are common challenges for stoma care in LMICs [10].

Anal dilatation is a distressing procedure for both infants and caregivers. Jenetzky et al. (2012) found that 69% of participants reported pain during anal dilatation on at least one occasion [34]. Caregivers were instructed to perform dilatations up to twice a day despite several studies in our review reporting that there is little evidence showing that dilatation can help prevent complications after surgery for ARM [34, 41–43]. These factors highlighted by our review require careful consideration so that, in the case that dilatations are prescribed, parents should have adequate guidance and support from healthcare professionals.

The Lancet Commission on Global Surgery highlights the lack of specialist surgical care facilities and personnel for people living in low-resourced settings [44]. The Commission reports that "44% of the world's population lives in countries with a specialist surgical workforce density lower than 20 per 100 000 population." [44]. This issue is particularly the case in LMICs where there is not only a surgical workforce deficit, but surgical facilities tend to be concentrated in urban areas. For those living at distances from hospitals where specialist surgical care is available, ongoing follow-up is especially challenging.

Our review found that most caregivers feel underprepared to deal with the wide array of care needs required by their infants post-discharge after surgery. These complex needs can be managed effectively at home, but caregivers need to be adequately prepared. Education pre-discharge should be prioritised so that caregivers feel empowered to meet these care needs. Education and training throughout the hospital stay and at subsequent intervals, has shown to increase information retention and knowledge in caregivers [35, 45]. Culturally appropriate and contextually relevant educational materials (videos, pamphlets etc.) may be provided upon discharge as these materials are effective educational strategies [46].

The role of community health workers (CHWs) for at-home care is increasingly recognised as effective in low-resource settings [47]. Support from CHWs may help relieve anxiety in parents, promote parental self-efficacy, and ultimately improve feeding and developmental outcomes in newborns [35, 47]. However, CHWs would require adequate training and remuneration to deliver this care.

Given the needs of these babies are complex, in addition to CHWs, a multidisciplinary team approach is essential to support both parents and infants [7, 48]. Studies in HICs have demonstrated the importance of a multidisciplinary team for neonatal surgical care [48]. Members of the care team would ideally include the surgeon, paediatrician, nurses, social workers, nutritionist, counsellors, and physical, occupational and speech therapists (when necessary) [48]. However, in resource-constrained settings, novel approaches to designing and delivering this level of multidisciplinary support are needed.

While the articles in our review covered a range of complex care needs, we identified several needs that were not referred to in any of the papers. These include wound care, pain relief, and dental care (to prevent endocarditis post cardiac surgery), and how to handle/lift infants post-surgery [49–51]. In our grey literature search we were able to find online materials from official sources (NHS, American Academy of Pediatrics) [49–51] to guide caregivers on the specific needs of their infants, but these were limited. Resources should be made readily available for caregivers post-discharge and cover guidance for the entire range of complex care needs.

## Need for context-appropriate solutions

Our review revealed that most studies discussing home care of newborns post-discharge take place in HICs. This is in keeping with studies by Chirdan et al. (2012) and Wright et al. (2021) that report a general lack of attention, funding, and support for research into neonatal surgical pathologies in LMICs [10, 52]. This is disproportionate given that 94% of congenital anomalies occur in these settings [3].

In their study exploring the mortality rate from gastrointestinal congenital anomalies, Wright et al. (2021) describe mortality rates of 39.8% in LICs [53]. This is compared to 20.4% in middle-income countries and 5.6% in HICs [53]. However, improved documentation and data systems, and further research, are required to reveal the true picture and explain reported mortality rates [10]. As recommended by the Lancet Commission on Global Surgery, more studies need to be done in low-resource settings to explore the specific challenges for caregivers, and tailor post-discharge educational materials [44].

## Limitations

A scoping review seeks to identify knowledge gaps, outline the range of studies, and synthesise evidence, using a narrative description. Quality appraisal, whilst important, is not a standard component of this methodology [8]. We chose a scoping review method given the heterogeneity of articles included and the fact that, for the purposes of our study (to map the scope of complex needs), a scoping review was an appropriate methodology [8].

Due to resource limitations, translation of non-English papers was not possible for this review. This is a limitation of our study, and we acknowledge that the inclusion of only English papers might exclude vital insights from non-English publications. This could affect the representation of studies from LMICs where English is not the primary language. The results of this work will be used to inform future research in non-English speaking settings to ensure an inclusive, holistic understanding of the post-discharge care needs of neonates in these communities.

Our review focused on a pre-defined set of congenital anomalies and is not inclusive of all congenital abnormalities that require surgery in the first year of life, including common conditions like cleft lip and palate. These conditions were chosen as they are common congenital surgical conditions that tend to result in death in the first year of life if left untreated and cover the most common congenital anomalies. These were also chosen in line the with the scientific literature and after consultation with clinicians working in the field.

Another important limitation is that this is a review of published articles, and so does not include the full picture of what is likely occurring in the homes of families of children with the full spectrum of congenital anomalies requiring surgery. Given this was a review of published papers, there is also publication bias towards cardiac disorders, like HLHS, which are uncommon, but due to the inherent nature of staged surgery for repair and palliation, have a component of post-discharge home monitoring, and so were overrepresented in our search strategy. This highlights the need for further research of all congenital abnormalities requiring surgery, especially cohort studies that explore experiences of families caring for babies requiring surgery beyond the immediate post-operative period.

## Conclusion

Congenital anomalies are one of the leading causes of death in children under 5 and 94% of these anomalies occur in LMICs [3]. The post-discharge period poses a challenge for caregivers who are expected to care for their infants after surgery. Our review highlighted that the complex care needs of these infants include monitoring, feeding, and specific care needs. Caregivers feel underprepared to care for their infants post-discharge. Given the array of care requirements, and the lack of access to specialised surgical facilities for those in LMICs, it is necessary to provide caregivers with the relevant education, training, and ongoing support at home.

Our scoping review highlights important aspects for the design and delivery of post-discharge care of vulnerable babies and their families. To fully understand the needs of those in

LMICs, future research which includes non-English populations not fully represented within our scoping review, is essential. This work is required to develop context-relevant strategies in these settings and ultimately contribute to lowering infant and under-five mortality.

## Supporting information

**S1 Checklist. Preferred Reporting Items for Systematic reviews and Meta-Analyses extension for Scoping Reviews (PRISMA-ScR) checklist.**
(PDF)

**S1 Table. Search strategy developed with University Librarian.**
(PDF)

**S2 Table. Data extraction table.** Please note references [45, 54–68] are included in the scoping review, further information on these papers is included in the data extraction table.
(PDF)

## Acknowledgments

The authors would like to acknowledge the essential help of Eli Harriss, Bodleian Health Care Libraries, University of Oxford, UK with the search strategy, choice of databases, and use of reference management software and systematic review software. We would also like to acknowledge the support of the Leo & Mia Foundation.

## Author Contributions

**Conceptualization:** Francesca Giulia Maraschin, Shobhana Nagraj.

**Formal analysis:** Francesca Giulia Maraschin, Fidelis Jacklyn Adella, Shobhana Nagraj.

**Methodology:** Francesca Giulia Maraschin, Fidelis Jacklyn Adella, Shobhana Nagraj.

**Supervision:** Shobhana Nagraj.

**Writing – original draft:** Francesca Giulia Maraschin.

**Writing – review & editing:** Francesca Giulia Maraschin, Fidelis Jacklyn Adella, Shobhana Nagraj.

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
