## [Decision Letter · Decision Letter 0]

23 Aug 2023

PGPH-D-23-01398

A scoping review of the post-discharge care needs of babies requiring surgery in the first year of life

Dear Dr. Maraschin,

Thank you for submitting your manuscript to PLOS Global Public Health. After careful consideration, we feel that it has merit but does not fully meet PLOS Global Public Health’s publication criteria as it currently stands. Therefore, we invite you to submit a revised version of the manuscript that addresses the points raised during the review process.

The major area that needs to be addressed surrounds the limitations of the study. Reviewer 2 in particular has highlighted that the lack of inclusion of non-English language studies is a major limitation of the manuscript, which also affects the conclusions made around a dearth of literature on the topic. If it is possible to expand the search to non-English language studies, this should be done. If it is not possible, then these limitations should be clearly delineated and addressed.

We look forward to receiving your revised manuscript.

Kind regards,

Shahrzad Joharifard, MD MPH

Academic Editor

Journal Requirements:

Additional Editor Comments (if provided):

The major area that needs to be addressed surrounds the limitations of the study. Reviewer 2 in particular has highlighted that the lack of inclusion of non-English language studies is a major limitation of the manuscript, which also affects the conclusions made around a dearth of literature on the topic. If it is possible to expand the search to non-English language studies, this should be done. If it is not possible, then these limitations should be clearly delineated and addressed.

Reviewers' comments:

Reviewer's Responses to Questions

**Comments to the Author**

1. Does this manuscript meet PLOS Global Public Health’s publication criteria? Is the manuscript technically sound, and do the data support the conclusions? The manuscript must describe methodologically and ethically rigorous research with conclusions that are appropriately drawn based on the data presented.

Reviewer #1: Yes

Reviewer #2: Yes

2. Has the statistical analysis been performed appropriately and rigorously?

Reviewer #1: Yes

Reviewer #2: Yes

3. Have the authors made all data underlying the findings in their manuscript fully available (please refer to the Data Availability Statement at the start of the manuscript PDF file)?

Reviewer #1: Yes

Reviewer #2: Yes

4. Is the manuscript presented in an intelligible fashion and written in standard English?

Reviewer #1: Yes

Reviewer #2: Yes

5. Review Comments to the Author

Reviewer #1: Congratulations on an excellent work.

Here are a few comments:

1. Add "systematic review" to the title instead of scoping?

2. Consider starting with discussing Monitoring then Feeding. As vital signs reflect a more critical baby than feeding.

3. Some conditions require special care were not mentioned as Cleft conditions

4. Consider adding a discussion about multidisciplinary teams/roles from surgeons, counselors, nurses and physiotherapists

Reviewer #2: Dear authors

I have had the opportunity to thoroughly review your study and would like to provide you with my feedback and insights.

Firstly, I want to commend you on addressing a crucial and often overlooked topic in healthcare. Your study sheds light on the complex care needs of newborns post-discharge, particularly focusing on congenital anomalies requiring surgery within the first year of life. This is a key area of research that holds significant implications for improving neonatal survival and caregiver support globally.

However, I have a few concerns and suggestions that I believe would enhance the quality and impact of your study:

1. Language and Inclusion of LMIC Studies:

While I understand the rationale behind restricting your search to English-language publications, it's essential to recognize that this approach may inadvertently marginalize vital research conducted in non-English languages, especially in low- and middle-income countries (LMICs). Most of the LMIC population and research does not use English as a primary language, and this linguistic disparity could result in overlooking valuable insights. In the context of your study's global surgery theme, it's imperative to acknowledge the limitations introduced by the language criterion and discuss how this could affect the representation of LMIC studies. Considering the importance of inclusive representation, I suggest reflecting upon this challenge in your manuscript and I would like to read your comments on that. Important nursing journals (as an example) coming from Latin American, Asia, and Africa were excluded in this review, and this is a major limitation. For future studies, I strongly suggest asking for help from other professionals/trainees that could help analyze papers in other languages.

2. Quality Assessment in Scoping Reviews:

While it's true that quality assessment is not a standard component of a scoping review, it's possible to do it. I would reframe the first sentence of the limitations section that affirms that quality assessment is not a part of scoping reviews.

3. Future Directions:

I would like to know how the presented study can contribute to the following research agenda. What should be the focus of future studies based on your results? And I do not agree with the author's conclusion that further studies in LMICs are needed as other languages were not considered. Maybe, there are plenty of studies there.

I would like to commend your clear writing style and the organization of your study. The identification of themes and sub-themes related to complex care needs was well-presented and effectively communicated. However, before any decision I would prefer to read the authors' response to the comments above.

Thank you for your time and effort in conducting this study. I look forward to seeing the continued progress of your research.

6. PLOS authors have the option to publish the peer review history of their article (what does this mean?). If published, this will include your full peer review and any attached files.

**Do you want your identity to be public for this peer review?** For information about this choice, including consent withdrawal, please see our Privacy Policy.

Reviewer #1: **Yes: **Mahmoud Elfiky

Reviewer #2: No

---

## [Decision Letter · Decision Letter 1]

2 Oct 2023

PGPH-D-23-01398R1

A systematic scoping review of the post-discharge care needs of babies requiring surgery in the first year of life

Dear Dr. Maraschin,

Thank you for submitting your manuscript to PLOS Global Public Health. After careful consideration, we feel that it has merit but does not fully meet PLOS Global Public Health’s publication criteria as it currently stands. Therefore, we invite you to submit a revised version of the manuscript that addresses the points raised during the review process.

We look forward to receiving your revised manuscript.

Kind regards,

Shahrzad Joharifard

Academic Editor

Journal Requirements:

Additional Editor Comments (if provided):

Thank you for making the suggested revisions. Both reviewers have suggested acceptance of your manuscript. Congratulations.

My only commentary is that this manuscript cannot be both a scoping review and a systematic review, so I take issue with the change that you have made to the title. Based on your methodology, it seems you have conducted a scoping review. If you believe otherwise, please justify this. If you agree that it is a scoping review, please make the change to the title and anywhere else in the manuscript where "systematic review" is mentioned.

Reviewers' comments:

Reviewer's Responses to Questions

**Comments to the Author**

1. If the authors have adequately addressed your comments raised in a previous round of review and you feel that this manuscript is now acceptable for publication, you may indicate that here to bypass the “Comments to the Author” section, enter your conflict of interest statement in the “Confidential to Editor” section, and submit your "Accept" recommendation.

Reviewer #1: All comments have been addressed

Reviewer #2: All comments have been addressed

2. Does this manuscript meet PLOS Global Public Health’s publication criteria? Is the manuscript technically sound, and do the data support the conclusions? The manuscript must describe methodologically and ethically rigorous research with conclusions that are appropriately drawn based on the data presented.

Reviewer #1: Yes

Reviewer #2: Yes

3. Has the statistical analysis been performed appropriately and rigorously?

Reviewer #1: Yes

Reviewer #2: Yes

4. Have the authors made all data underlying the findings in their manuscript fully available (please refer to the Data Availability Statement at the start of the manuscript PDF file)?

Reviewer #1: Yes

Reviewer #2: Yes

5. Is the manuscript presented in an intelligible fashion and written in standard English?

Reviewer #1: Yes

Reviewer #2: Yes

6. Review Comments to the Author

Reviewer #1: Thank you for making the necessary changes. As a surgeon and a coauthor to one of the references, I am really ecstatic with the changes and work ofcourse

Reviewer #2: The authors have addressed all my comments. Thank you and congratulations.

7. PLOS authors have the option to publish the peer review history of their article (what does this mean?). If published, this will include your full peer review and any attached files.

**Do you want your identity to be public for this peer review?** For information about this choice, including consent withdrawal, please see our Privacy Policy.

Reviewer #1: **Yes: **Mahmoud Elfiky

Reviewer #2: No

---

## [Decision Letter · Decision Letter 2]

2 Nov 2023

A scoping review of the post-discharge care needs of babies requiring surgery in the first year of life

PGPH-D-23-01398R2

Dear Dr Maraschin,

We are pleased to inform you that your manuscript 'A scoping review of the post-discharge care needs of babies requiring surgery in the first year of life' has been provisionally accepted for publication in PLOS Global Public Health.

Best regards,

Julia Robinson

Executive Editor

Reviewer Comments (if any, and for reference):

Reviewer's Responses to Questions

**Comments to the Author**

1. If the authors have adequately addressed your comments raised in a previous round of review and you feel that this manuscript is now acceptable for publication, you may indicate that here to bypass the “Comments to the Author” section, enter your conflict of interest statement in the “Confidential to Editor” section, and submit your "Accept" recommendation.

Reviewer #1: All comments have been addressed

Reviewer #2: All comments have been addressed

2. Does this manuscript meet PLOS Global Public Health’s publication criteria? Is the manuscript technically sound, and do the data support the conclusions? The manuscript must describe methodologically and ethically rigorous research with conclusions that are appropriately drawn based on the data presented.

Reviewer #1: Yes

Reviewer #2: Yes

3. Has the statistical analysis been performed appropriately and rigorously?

Reviewer #1: Yes

Reviewer #2: Yes

4. Have the authors made all data underlying the findings in their manuscript fully available (please refer to the Data Availability Statement at the start of the manuscript PDF file)?

Reviewer #1: Yes

Reviewer #2: Yes

5. Is the manuscript presented in an intelligible fashion and written in standard English?

Reviewer #1: Yes

Reviewer #2: Yes

6. Review Comments to the Author

Reviewer #1: Thank you for responding to all comments. This manuscript is a great addition to literature

Reviewer #2: No additional comments.

7. PLOS authors have the option to publish the peer review history of their article (what does this mean?). If published, this will include your full peer review and any attached files.

**Do you want your identity to be public for this peer review?** For information about this choice, including consent withdrawal, please see our Privacy Policy.

Reviewer #1: **Yes: **Mahmoud Elfiky

Reviewer #2: No
